# Sexual Functioning in Pregnant Women

**DOI:** 10.3390/ijerph16214216

**Published:** 2019-10-30

**Authors:** Anna Fuchs, Iwona Czech, Jerzy Sikora, Piotr Fuchs, Miłosz Lorek, Violetta Skrzypulec-Plinta, Agnieszka Drosdzol-Cop

**Affiliations:** 1Department of Pregnancy Pathology, School of Health Sciences in Katowice, Medical University of Silesia, 40-752 Katowice, Poland; iwonaczech232@gmail.com (I.C.); jerzy_sikora@poczta.onet.pl (J.S.); revisfux@gmail.com (P.F.); lorek.milosz@gmail.com (M.L.); cor111@poczta.onet.pl (A.D.-C.); 2Department of Woman’s Health, School of Health Sciences in Katowice, Medical University of Silesia, 40-752 Katowice, Poland; skrzypulec-plinta@o2.pl

**Keywords:** sexuality, pregnancy, intercourse, desire, sexual activity in pregnancy, pain

## Abstract

Sexual activity during pregnancy is determined by emotional, psychosocial, hormonal, and anatomical factors and varies during trimesters. This work aimed to establish women’s sexual activity during each trimester of pregnancy. A total of 624 women were included in the study and filled in the questionnaire three times, once during each trimester of pregnancy. The first part of the survey included questions about socio-demographic characteristics, obstetric history, and medical details of a given pregnancy. The second part was the Polish version of the female sexual function index (FSFI) questionnaire. Comparison of the mean scores for the overall sexual function of each trimester revealed clinically relevant sexual dysfunction in the second and third trimesters (mean values 25.9 ± 8.7 and 22.7 ± 8.7, respectively; *p* < 0.01). Women were most sexually active during their second trimester. In the first trimester of pregnancy, women were most likely to choose intercourse in the missionary position. Women with vocational education were characterized by the lowest and homogenous FSFI values. Total FSFI score depended on the martial status—the highest value pertained to married women (25.2 ± 6.9; *p* = 0.02).

## 1. Introduction

Sexuality is an important part of human life, one that crucially determines people’s well-being. As pregnancy is one of the most important periods in one’s life, it may be a cause of an alteration in the sexual activity of future parents. Pregnancy is often associated with a reduction or cessation of sexual activity [1]. This may be a result of decreased sexual desire that is often reported by pregnant women, or it is caused by the fears and myths associated with the health of both mother and child [2,3]. The most often mentioned concerns about sexual activity during pregnancy are bleeding, induction of labor, infection, fetal damage, and the rupture of membranes [1,3]. It must also be highlighted that women who are satisfied with their sex life may be interested in the continuation of sexual activity throughout their whole pregnancy, while those who are not interested in sex may tend to avoid sexual intercourse during pregnancy. 

Several scales are used to measure sexual functioning, such as the sexual interaction inventory, the Arizona sexual experiences scale, the sexual satisfaction questionnaire, or the female sexual function index (FSFI). The last-mentioned is the one that is most frequently used for this purpose. It evaluates a woman’s sexual function in six domains, namely: desire, arousal, lubrication, orgasm, satisfaction, and pain. 

Sexual behavior during pregnancy is determined by emotional, hormonal, psychosocial, and anatomical factors and varies during each trimester of pregnancy [4]. What is interesting is the fact that the frequency of sexual activity during one’s whole pregnancy also varies between primiparae and multiparae. The first group tends to have sexual intercourse less frequently than before the pregnancy, while there are no such differences in the multiparae group [5]. 

### 1.1. First Trimester

The first trimester, referred to as the period of adaptation, lasts from conception to the twelfth week of pregnancy. The female body must adapt to neurohormonal changes that are responsible for reducing one’s physical performance, and inducing drowsiness, vomiting, or mood swings. There are the greatest fluctuations concerning the frequency of sexual intercourse, from normal sexual activity to a total cessation of any form of sexual activity during this trimester. Fok et al., in their cross-sectional study conducted in Hong Kong, established that more than one-third of women stopped vaginal intercourse during pregnancy. This is probably because of the myths that vaginal intercourse during pregnancy may cause miscarriage, premature labor, or fetal damage. In the same study, results show that 55.6% of women in the first trimester performed vaginal intercourse [1].

What is interesting is the fact that Corbaciolgu et al. observed that women who were unaware of their pregnancy in early gestation had a significantly higher frequency of sexual intercourse than those who were aware of their pregnancy [6].

### 1.2. Second Trimester

In the second trimester of pregnancy, which lasts from 13 to 26 weeks, the frequency of intercourse usually increases [3]. Most researchers claim that sexual function improves during the second trimester of pregnancy when compared to the first trimester [7]. However, some researchers have found that function may not change during the second trimester of pregnancy when compared to the first trimester [7,8,9]. This is connected with a greater interest in sexuality, a reduction in the physical symptoms of pregnancy, feeling better, and having greater self-confidence [9]. The hyperemia of the reproductive organs and the more intensive wetting of the vaginal walls have a great influence on this. Because of vascular changes in the vagina and vulva, satisfaction may be greater than before pregnancy [9]. 

### 1.3. Third Trimester

The last stage of pregnancy, lasting from week 27 to delivery, is characterized by the lowest frequency of sexual activity. The lowest level of libido was observed in women when compared with the previous trimesters of pregnancy [10]. During the third trimester, as many as 52% to 73% of women meet clinical cutoffs on standardized measurements for sexual problems [3]. The third trimester is psychologically the most difficult one for the future mother, as it is associated with anxiety and stress concerning labor and motherhood. Because of the fear of inducing labor or harming the child, many couples decide to stop conducting sexual activity altogether. In addition, the changes in a woman’s body shape and in her anatomy may impede a couple performing sexual intercourse. Erol et al. observed that the FSFI score was decreased in the third trimester in comparison with both previous trimesters [11]. The impact on the frequency of sexual intercourse may have emerging somatic changes, such as swelling, fatigue, increased body mass, increased vaginal discharge, as well as mental exhaustion. Many couples may be afraid of inducing preterm delivery. Restriction of sexual intercourse is routinely recommended for the prevention and management of threatened preterm labor because of the theoretical risk of intercourse as a method of inducing labor. However, the existing literature is contradictory and limited by study design, reporting bias, and the rarity of preterm labor as an event [12,13].

This work aimed to establish the sexual activity of pregnant women during each trimester of pregnancy. Little is known about the sexual functioning of women in Poland, and this is the first large study investigating this topic in pregnant Silesian women. 

## 2. Material and Methods

### 2.1. Study Group

Our study sample was derived from the Department of Pregnancy Pathology, the Department of Woman’s Health in the School of Health Sciences at the Medical University of Silesia, Katowice, Poland, between January 2017 and January 2018. Pregnant women (n = 726) were recruited personally after the pregnancy was confirmed by transvaginal ultrasound and screened for inclusion and exclusion criteria. Gestational age was established from the last menstrual cycle and verified by ultrasound scan measurements. Eligible women who were healthy, pregnant, aged eighteen or older, with no aggravating medical history, and who gave written and informed consent to their participation were included in the study. Exclusion criteria were a history of miscarriages, suspicion of congenital defects, threatened abortion, placenta previa, cervical incompetence, intrauterine growth restriction multiple pregnancies, pregnancy after assisted reproductive technology, absence of sexual partner/lack of sexual activity in previous four weeks, and lack of informed consent of the patient.

Women completed the self-administered questionnaire once in each trimester of pregnancy during consecutive checkups or childbirth classes: (T1) between 7th to 12th week of gestation, (T2) between 13rd to 24th week of gestation, (T3) between 25th to 39th week of gestation. 

Ninety-six patients (94.12%) were also excluded while conducting the study due to lack of sexual activity during the third trimester, and finally, 624 women were included in the study.

The university Ethics Committee waived the requirement for informed consent due to the anonymous and non-interventional nature of the study (KNW/0022/KB/68/19).

### 2.2. Questionnaire

The survey consisted of two parts. The first part included questions regarding the socio-demographic characteristics, obstetric history, and medical details of the pregnancy concerned. The second part was the Polish version of the female sexual function index (FSFI) questionnaire. The FSFI is a validated questionnaire containing 19 questions and measures a women’s sexual functioning across six domains: desire (questions 1 and 2; score range 1–5); arousal (questions 3,4,5,6; score range 0–5); lubrication (questions 7, 8, 9, 10; score range 0–5); orgasm (questions 11, 12, 13; score range 0–5); satisfaction (questions 14, 15, 16; score range 1–5); and pain (questions 17, 18, 19; score range 0–5) during the previous 4 weeks. The full-scale score range is 2 to 36, with lower scores being associated with worse sexual function. In addition, in the second part of the questionnaire, women were asked to report their concerns associated with sexual intercourse in subsequent trimesters and the frequency of intercourse. 

The questionnaire was first pretested on a group of 20 women who were asked to comment on the clarity of the questions. 

### 2.3. Statistical Analysis

All data analyses were conducted using StatSoft Statistica version 13.0 PL software (Dell, TX, USA), and a *p*-value < 0.05 was considered as significant. As it is adjusted for dependent variables, the Friedman ANOVA test was performed to compare the results of subsequent trimesters, while the Wilcoxon’s rank test was applied as a post hoc analysis. For the sake of quantitative variables, comparison Chi^2 was utilized. Kruskal–Wallis test was implemented for independent variables, and further investigations were accomplished by the U Mann–Whitney test. Data are presented as a mean ± SE (Standard Error).

## 3. Results

The subjects’ characteristics are presented in Table 1. The mean age of the participants was 28.2 ± 5.5 years. A total of 59.1% of patients were less than 30 years old. Moreover, 344 women were married (55.1%), and 280 (44.9%) were in an informal relationship. None of the patients was single, while 47.8% (n = 298) were nulliparous, and 52.2% (n = 326) were undergoing their second or subsequent pregnancy. In addition, 88% (n = 554) did not report any complications during the course of the pregnancy. The most common problems among patients included ere nausea and emesis (n = 170; 27.2%); urinary tract infection (n = 104; 16.7%), and spotting blood (n = 48; 7.7%). A total of 296 patients (47.4%) had college or university degree, 166 (26.6%) were high school graduates, 96 (15.4%) had a vocational education, while 66 (10.6%) women had not graduated from any school. Moreover, 142 (22.8%) patients resided in a large city of over 250,000 residents, 278 (44.6%) were residents of a city with a population of 50,000 to 250,000, while 116 (18.6%) patients lived in towns and 88 (14.1%) in villages. 

The average FSFI score results during the three trimesters are presented in Table 2. Comparison of the mean scores for overall sexual function of each trimester revealed clinically relevant sexual dysfunction in the second and third trimesters (mean values 25.9 ± 8.7 and 22.7 ± 8.7, respectively). However, there was no difference between the first and the second trimesters (*p* > 0.05), while there was a significant decrease in the third trimester compared with both the first and second trimesters (*p* < 0.01).

Mean overall FSFI equaled 24.7 ± 6.7 and did not correlate with age and parity (both *p* > 0.05). Total score depended on the martial status—the highest value pertained to married women (25.2 ± 6.9) and differed from women with an informal relationship (24.5 ± 6.0; *p* = 0.02). Women with vocational education were characterized by the lowest and homogenous FSFI values (23.6 ± 4.4) and differed from women with higher education (25.2 ± 7.4; *p* ≤ 0.001) and from women with secondary education (25.1 ± 3.5; *p* = 0.03, Figure 1). Place of residence, along attendance at parental craft classes, did not affect sexual functioning (*p* = 0.09 and *p* = 0.34, respectively). 

Women were the most sexually active during their second trimester (*p* < 0.05), with a median of eight intercourses per month. The median number of intercourses in the first and third trimester was four intercourses per month.

In the first trimester of pregnancy, women were most likely to choose intercourse in the missionary position (*p* < 0.01). In the second and third trimester, the most-chosen position was the spoons position (*p* > 0.05). Other positions most often mentioned by the patients were cowgirl and doggy style positions (Figure 2). 

Out of 102 patients who were excluded from the study, 96 patients (94.12%) claimed that they were sexually inactive during the third trimester of pregnancy, while there was no statistically significant difference (*p* > 0.05) between first and second trimesters (33 and 44, respectively) (Figure 2).

## 4. Discussion

Nowadays, there is a growing interest in the topic of the sexuality of pregnant women. There is growing awareness among people that sexuality does matter, while sexual activity has ceased to be a taboo subject. Moreover, while in the past, pregnant women often felt unattractive, now they are eager to celebrate pregnancy and are proud of it.

In our study, we established the change in frequency of sexual intercourse throughout pregnancy and, in addition, the variations in sexual desire, arousal, lubrication, pain, orgasm, satisfaction between trimesters. We decided to use the FSFI questionnaire as it is the survey used worldwide and has been employed in many previous studies. Therefore, it is a reliable tool and, what is more, has a validated Polish version.

Serati et al. analyzed literature published from 1960 to 2010 and observed that most of the studies show a significant decline in female sexual function and the frequency of sexual activity during pregnancy [5]. Apart from this, the authors of a cross-sectional observational study conducted in Hong Kong observed that inadequate knowledge, cultural factors, and anxiety are the aspects associated with the reduction of sexual activity [1]. Moreover, physiological and anatomical changes that occur during pregnancy may have an impact as well. Anatomical factors affecting sexual activity during pregnancy are

Enlargement of the abdominal girth;

Breast enlargement;

Discharge;

Skin pigmentation;

Limb edema;

Congestion and increased vaginal humidity;

Varicose veins.

### 4.1. First Trimester

The first trimester is connected with a diminution of sexual intercourse frequency due to nausea, sickness, breast sensitivity, and a worsening sense of well-being [4,7]. 

Corbaciolgu-Esmer et al., in their study in Turkey, showed that almost 60% of the surveyed patients reported a decrease in the frequency of sexual intercourse in the first trimester [14]. In our study, there was no statistically significant difference in overall sexual functioning between the first and second trimesters. Our finding is in agreement with the results of an American study carried out at the University of New Mexico [7].

Corbacioglu-Esmer et al. also reported that between the first two trimesters, there are no significant changes in any FSFI domains [14]. As already mentioned, our study confirms this report. 

British studies report that sexual intercourse was also associated with reduced odds of miscarriage unless there was bleeding during intercourse, in which case, the odds of miscarriage almost doubled. What is more, the authors clearly state that sexual intercourse during the first trimester is one of the most important indicators of well-being [15].

It should be remembered, however, that if any pathology of pregnancy is diagnosed in the first trimester, sexual activity should be limited [2].

### 4.2. Second Trimester

In the second trimester, the frequency of sexual intercourse rises, which is connected with a higher sense of security and increasing sexual interest. In addition, the smaller number of physical symptoms connected with pregnancy than in the previous trimester, ameliorate a woman’s sense of well-being [5,8]. In our study, during their whole pregnancy, women were most sexually active in the second trimester (a median of eight incidences of intercourse per month). 

Aslan et al. found that the sexual functions in all domains of the FSFI declined in the second trimester when compared with the first trimester [10]. Their statement is consistent with Bartellas’s findings [16]. Our findings show a diminution in scores both in the pain and satisfaction domains. Higher scores were observed in the desire and arousal domains. The score in the orgasm domain did not vary between the first and the second trimester, which is in agreement with Ninivaggio’s results [7].

### 4.3. Third Trimester

Corbacioglu-Esmer et al. reported significantly lower sexual function scores in the third trimester in comparison with the first two trimesters [14].

We observed a decline in all domains of the FSFI in the last trimester. This finding is in agreement with other reports that show lower scores of FSFI domains in the third trimester [10]. On the other hand, Chang et al. observed a statistically significant increase in sexual satisfaction in the third trimester, although this is the only study with such results [4].

In our study, the biggest decrease in the score was associated with the pain domain, which means that pregnant women in the third trimester of pregnancy were suffering from pain during sexual relations statistically significantly more than in previous trimesters. What is more, the third trimester was associated with a reduction in the frequency of sexual activity. This is probably due to anatomical changes that affect the pregnant body and concerns about the health of the child.

The third trimester of pregnancy is the most difficult for a woman for psychological and anatomical reasons. During this period, special care should be taken in this regard. In a study carried out in Canada, only one-third of women received an education from their provider about sexual activity during pregnancy, and nearly half brought up the topic themselves [16]. One should also help pregnant women in choosing the right sexual positions. According to our study, women in the third trimester of pregnancy prefer the spoons position.

The overall FSFI results were below the reference interval, which may indicate that pregnant women are at risk of developing sexual disorders during pregnancy. According to our study, the most significant factor that increases the possibility of sexual dysfunctions is a low educational level. Our study remains consistent in this matter with Banaei et al., who also found out that sexual dysfunction was lower in women with a higher level of education [17]. Our finding that women with lower educational levels were more likely to report disturbed sexual function is also in agreement with the results obtained by Abouzari-Gazafroodi [2]. Therefore, it is important to educate future parents, that in cases of physiological pregnancy, there are no contraindications regarding conducting sexual activity. In fact, satisfying sexual activity during pregnancy plays a crucial role in maintaining a proper, pleasant relationship between future parents [18]. It is a natural step forward in a couple’s life. Many couples lack reliable and adequate knowledge about sex during pregnancy, a phenomenon that results in them avoiding sexual activity during this period [5,19]. Obstetricians should provide practical advice concerning sexual activity during pregnancy and eliminate fears of future mothers. In cases of indications for ceasing sexual intercourse, an obstetrician should help couples in finding alternative ways of conducting intimate contact [18].

The limitation of our study is a lack of information about the sexual functioning of expectant fathers. It is also important to examine their sexual desire and satisfaction throughout pregnancy. In our future research, we will focus on this aspect of sexual experiences during pregnancy. However, we have to emphasize that the large study group provides more reliable evidences. The prospective character of our research allows us to investigate sexual functions among the same sample during all following trimesters. 

## 5. Conclusions

We observed a statistically significant decrease in sexual functioning in women during the third trimester of pregnancy when compared with the first and second trimesters. These women should be provided with practical advice concerning sexual activity. We observed different preferred sexual positions during each trimester of pregnancy. This observation may be used by sexologists and gynecologists while advising future parents on the subject of their sexual activity. 

## Figures and Tables

**Figure 1 ijerph-16-04216-f001:**
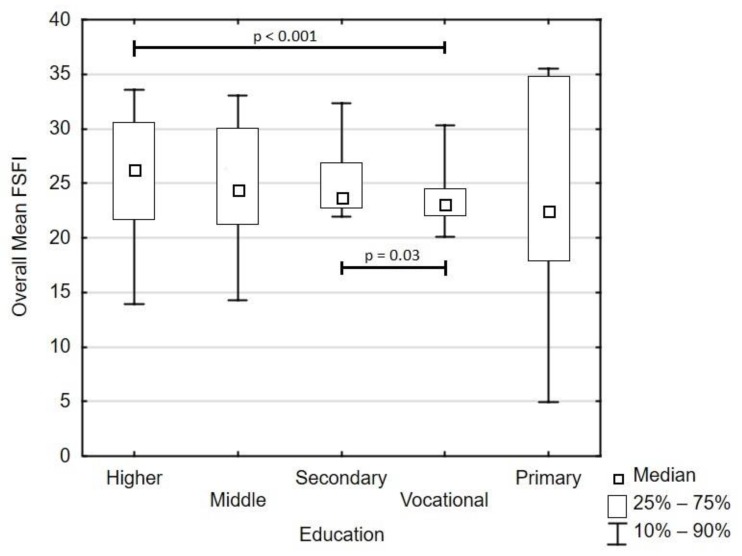
Comparison of overall female sexual function index (FSFI) depending on educational level.

**Figure 2 ijerph-16-04216-f002:**
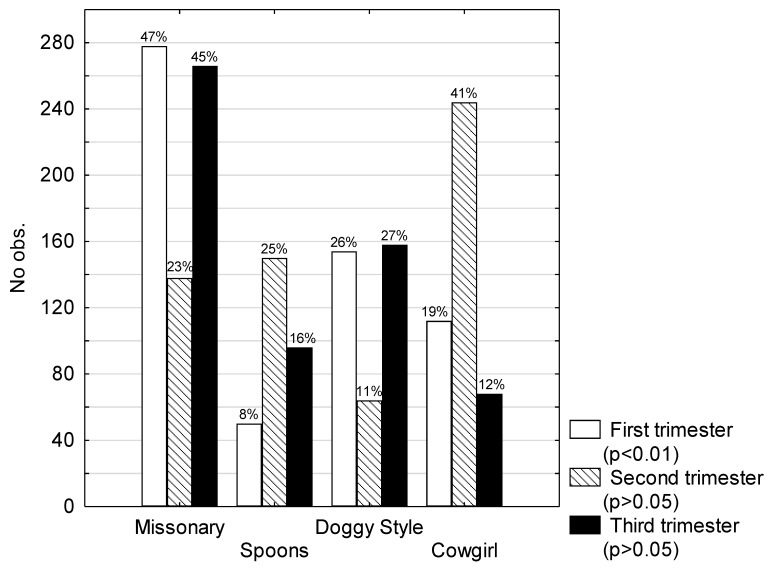
Sex positions chosen by pregnant patients.

**Table 1 ijerph-16-04216-t001:** Subjects’ characteristics.

Participant Characteristics	No (%)
**Age (years)**	624
18–30	364 (59.1)
Over 30	252 (40.9)
**Marital status**	624
Married	344 (55.1)
Informal relationship	280 (44.9)
**Parity**	624
Primiparity	298 (47.8)
Multiparity	326 (52.2)
**Non-complicated (physiological) pregnancy**	624
Yes	554 (88.8)
No	70 (11.2)
**Complications during pregnancy**	382
Nausea and emesis	170 (44.5)
Urinary tract infections	104 (27.2)
Spotting blood	48 (12.6)
Gestational diabetes	32 (8.4)
Hypertension	26 (6.8)
Varicose veins	2 (0.5)
**Education**	624
College or University Degree	296 (47.4)
High School	166 (26.6)
Vocational	96 (15.4)
No education	66 (10.6)
**Place of residence**	624
City above 250,000 residents	142 (22.8)
City 50,000–250,000 residents	278 (44.6)
Town below 50,000 residents	116 (18.6)
Village	88 (14.1)

**Table 2 ijerph-16-04216-t002:** Assessment of sexual functioning in subsequent trimesters of pregnancy.

Variables	First Trimester	Second Trimester	Third Trimester	*p*
Sexual Intercourse *	4 (1;16)	8 (1;16)	4 (4;12)	<0.01
FSFI domain				
Desire	3.9 ± 1.1	4.2 ± 1.2	3.8 ± 1.4	<0.01
Arousal	4.4 ± 1.1	4.5 ± 1.3	3.8 ± 1.6	<0.01
Lubrication	4.5 ± 1.4	4.3 ± 1.7	3.8 ± 1.9	<0.01
Orgasm	4.3 ± 1.4	4.3 ± 1.7	3.7 ± 1.9	<0.01
Satisfaction	4.7 ± 1.2	4.6 ± 1.3	4.3 ± 1.4	<0.01
Pain	4.2 ± 1.6	4.0 ± 1.8	3.5 ± 2.0	<0.01
Total	26.1 ± 6.1	25.9 ± 8.7	22.7 ± 8.7	<0.01

* per month (in median (10th, 90th percentiles)) Female Sexual Function Index (FSFI)

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
