# Peer review of "Sexual Functioning in Pregnant Women"

_ijerph, 2019, doi:10.3390/ijerph16214216_

Round 1

Reviewer 1 Report

In my opinion the manuscript can be accepted for publication in IJERPH. I know that such research exists but there are no studies on this topic in Poland.

Some minor comments:

Please add p values (when appropriate) into the abstract of the paper One sentence requires correction “The impact on the frequency of sexual intercourse may have emerging somatic changes such as: cross-bones [unclear??]” Table 1 can be included in supplementary materials (I would delete the last column with means) Figure 2 requires improvement (please correct the colors - that included in the legend do not correspond with the picture (for 3rd trimester) – you can consider percentages instead the numbers.

Author Response

Dear Reviewer,

Thank you for your valuable comments. Please let us show you, step by step, how we changed our manuscript according to your suggestions:

Please add p values (when appropriate) into the abstract of the paper.

We did it.

One sentence requires correction “The impact on the frequency of sexual intercourse may have emerging somatic changes such as: cross-bones [unclear??]”

Indeed, it was an unfortunate statement, we changed it .

Table 1 can be included in supplementary materials (I would delete the last column with means)

We deleted the last column with means and we also will include this table in supplementary materials.

Figure 2 requires improvement (please correct the colors - that included in the legend do not correspond with the picture (for 3rd trimester) – you can consider percentages instead the numbers.

We changed it.

Our best regards,

The Authors

Reviewer 2 Report

Thank you for the opportunity to review your paper. Your research is important and of high interest.

What means "Normal pregnancy" in table 1.?

Please add p-value to Figure 1 and 2.

Author Response

Dear Reviewer,

Thank you for your valuable comments. Please let us show you, step-by-step, how we changed our manuscript according to your suggestions:

What means "Normal pregnancy" in table 1.?

Indeed, it was an unfortunate statement, we changed it to "Non-complicated (physiological) pregnancy".

Please add p-value to Figure 1 and 2.

We have done it.

Our best regards,

The Authors

This manuscript is a resubmission of an earlier submission. The following is a list of the peer review reports and author responses from that submission.

Round 1

Reviewer 1 Report

Thank you for giving me the opportunity to review the manuscript entitled “ What happens in bed during the trimesters of pregnancy? Sexual activity in Silesian pregnant women.”

Please find below several comments which in my opinion can improve the paper:

The firs part of the title looks very strange and does not correspond to the scientific paper – it needs to be rewritten. Maybe you can leave the second part of the title only. The first affiliation contains a mistake “ De and rtment” Page 1 line 18 space needs to be added The aim of the study needs to be improved. I would leave only the first part of the sentence “The aims of this work was to establish women’s sexual activity during each trimester of pregnancy.” What is the added value of the next part of the sentence describing the aim? I would just present the general aim of the study. I would not use “pregnants” better “pregnant women” I would change the sentence “….self-administered questionnaire three times, during each trimester of pregnancy during consecutive checkups or childbirth classes…..” into “self-administered questionnaire once in each trimester of pregnancy during consecutive checkups or childbirth classes….” Was the study approved by Bioethical committee? The socio-demographic and medical variables need to be briefly desribed. In the results part of the manuscript there is no need to present the numbers, percentages are sufficient. "Absence of sexual partner/lack of sexual activity in previous four weeks" was described as exclusion criteria so how it is related to “I was not active sexually during this trimester” (which is also presented for the 1st trimester) and other analyzed variables? It is strange that “None of the patients were single.” What is understood by single (not married or not having partner)? I would expect some women being single. How it can be explained? Or maybe it is because of exclusion criteria “absence of sexual partner/lack of sexual activity in previous four weeks”. In the sentence “Nonetheless, there was no significant difference between the first and the second trimester (p>0.05) and a significant decrease in third trimester (p<0.01) was observed.” - it means decrease in third trimester comparing to the second trimester? This needs to be added. The fig. 1 is not adding new information (it needs to be deleted). Sexual Intercourses per month are presented in table 2. p-values can be added into fig. 2 Data presented in fig. 3 can be added into table 1 (and presented as n and %) – so the fig. 3 can be deleted. In that case relevant text needs to be moved earlier. Additional analysis can be done looking at the impact of socio-demographic and medical variables on sexual activity during pregnancy. Sometimes is written Lew-Starowicz and sometimes Lew Starowicz – it needs to be uniform.

Reviewer 2 Report

Introduction

Line 33: Why is pregnancy "one of the most critical periods"

Line 52: Please explain why the first trimester is referred to as "mental crisis"

Line 64-65: Please provide citation for this claim

Line 65-66: Please provide citation for this claim

Line 67: "thanks to vascular changes..." this sentence is unclear, please re-write to convey what is meant here. 

Line 74-75: Please provide a citation for this claim. 

In sections 1.1. to 1.3 strong claims are made based on just one or two studies (3 and 4 are not in english so I am not able to review these). I suggest rewriting these claims to more accurately reflect the samples of these studies and not generally claim "men"/"women" experience X or Y. 

The aims of this work were to: Establish women’s sexual activity during each trimester of pregnancy . Assess how women’s sexual activity changes during pregnancy. Determine the frequency of sexual intercourses during pregnancy. However, the Introduction lacks any argument as to why these aims are significant and needed to be studied aside from the mention that pregnancy is when couples can expect sexual fluctuations. I do not find this to be a compelling argument for the aims mentioned. Further, the review of literature needs significant improvement via more details on the studies cited and how these studies define sexual activity and their samples and data collection points. 

Methods:

Define "recruited personally"

Please provide justification for exclusion criteria listed

Line 105: previous studies have found significant difference in sexual activity frequency between early and late third trimester--since this study does not discriminate between early and late third tri, this should be included in the limitations later in the manuscript

If sexual activity is operationalized as vaginal intercourse than the manuscript should be fully edited to only discuss vaginal intercourse and not generalize to sexual activity. This includes data pulled from previous studies should only be limit4ed to vaginal intercourse frequency 

Results and Discussion cannot be reviewed thoroughly until the above details are taken care of. 

Reviewer 3 Report

Sexuality is an important component of health and well‐being in a woman's life. The current manuscript investigates how women's sexual activity changes during pregnancy and determine the frequency of sexual intercourse during pregnancy. There is a large body of evidence that addresses the concept investigated in this study, and hence the findings are not novel and provide limited value addition to our existing knowledge.

Reviewer 4 Report

Interesting study, however, still requires a few corrections.
Please cite works from 2018 and 2019 in the article. On 21 items reference one is from 2017, the others are older.

lines 41-48 -
Information on the scale used for the tests should be found in Materials and methods, while an explanation of why this scale was used should be included in the discussion. Attaching BMI calculations to the results could be interesting.

lines 162-165 - please provide references;
Expand sentence: "There are publications appearing." The conclusions should be completely changed because they do not result from the presented research.

Round 2

Reviewer 1 Report

In my opinion this manuscript can be accepted for publications. The authors have followed my comments.

Reviewer 3 Report

The authors have suitably addressed all the concerns raised in the revised manuscript and can be accepted for publication in your esteemed journal.